# Mixed-methods education of mechanical ventilation for residents in the era of the COVID-19 pandemic: Preliminary interventional study

Kenichiro Takeda[1,2]*, Hajime Kasai[1,2,3], Hiroshi Tajima[1,2,3], Yutaka Furukawa[4], Taro Imaeda[2,5], Takuji Suzuki[1], Shoichi Ito[2,3]

1 Department of Respirology, Graduate School of Medicine, Chiba University, Chiba, Japan, 2 Health Professional Development Center, Chiba University Hospital, Chiba, Japan, 3 Department of Medical Education, Graduate School of Medicine, Chiba University, Chiba, Japan, 4 Clinical Engineering Center, Chiba University Hospital, Chiba, Japan, 5 Department of Emergency and Critical Care Medicine, Graduate School of Medicine, Chiba University, Chiba, Japan

* k.takeda@chiba-u.jp

## Abstract

### Introduction

In the current era of the severe acute respiratory syndrome-coronavirus-2 epidemic, the need for respiratory care, including mechanical ventilatory (MV) management, has increased. However, there are no well-developed educational strategies for training medical personnel dealing with respiratory care in MV management.

### Methods

A novel mixed-methods hands-on seminar for learning MV management was conducted for the residents at Chiba University Hospital in March 2022. The seminar lasted approximately 2 hours. The learning goal for the residents was to develop skills and knowledge in performing basic respiratory care, including MV, during an outbreak of a respiratory infection. The seminar with a flipped classroom consisted of e-learning, including modules on respiratory physiology and MV management, hands-on training with a low-fidelity simulator (a lung simulator), and hands-on training with a high-fidelity simulator (a human patient simulator). The effectiveness of the seminar was evaluated using closed questions (scored on a five-point Likert scale: 1 [minimum] to 5 [maximum]) and multiple-choice questions (maximum score: 6) at the pre- and post-seminar evaluations.

### Results

Fourteen residents at Chiba University Hospital participated in the program. The questionnaire responses revealed that the participants' motivation for learning about MV was relatively high in the pre-seminar period (seven participants [50%] selected level 5 [very strong]), and it increased in the post-seminar period (all participants selected level 5) ($p$ = 0.016). The responses to the multiple-choice questions revealed that the participants did not

**Data Availability Statement:** All relevant data are within the manuscript and its Supporting Information files.

**Funding:** The authors received no specific funding for this work.

**Competing interests:** The authors have declared that no competing interests exist.

**Abbreviations:** A/C, assist/control; CPAP, continuous positive airway pressure; ETCO$_2$, end tidal CO$_2$; FGIs, focus group interviews; F$_I$O$_2$, fraction of inspiratory oxygen; MCQs, multiple-choice questions; MV, mechanical ventilation; PEEP, positive end-expiratory pressure.

have enough knowledge to operate a mechanical ventilator, while the total score significantly improved from the pre- to post-seminar period (pre-seminar: 3.3 ± 1.1, post-seminar: 4.6 ± 1.0, $p$ = 0.003).

## Conclusions

The seminar implemented in this study helped increase the residents' motivation to learn about respiratory care and improved knowledge of MV management in a short time. In particular, the flipped classroom may promote the efficiency of education on MV management.

## Introduction

Since December 2019, the coronavirus disease 2019 (COVID-19) has been spreading rapidly on a global scale, resulting in a major impact on healthcare services [1]. COVID-19 often develops into pneumonia and severe respiratory failure, requiring oxygen therapy and/or mechanical ventilation (MV) [2]. Patients with severe respiratory failure and COVID-19 pneumonia usually need to be treated by well-trained professionals and require respiratory care in advanced medical institutions. In hindsight, other types of respiratory infections that cause severe respiratory failure have broken out regularly throughout the world or in specific regions [3,4]. Therefore, training the medical personnel responsible for respiratory care is an urgent requirement for health services both in Japan and the global community [5] in preparation for epidemics of infectious diseases, other than COVID-19, in the future.

Standardized educational strategies for MV management have not yet been established [6]. In fact, a report from the United States, published in 2003, indicated that the majority of internal medicine residents were dissatisfied with the ventilatory management education they had received [7]. Therefore, traditional on-the-job training is not sufficient as an educational method for MV management. Hands-on simulation-based education for MV, introduced in the last ten years, has shown high participant satisfaction [8–14]. However, the content included in the hands-on training in previous studies varied greatly, and there is no standardized strategy, although high-fidelity simulations with human patient simulators are often used. Additionally, in previous studies, the average duration of a hands-on simulation-based training was approximately seven hours [6] and engaging in training for such long durations can be challenging for both learners and instructors. Additionally, many intensive care unit staff do not receive sufficient MV education [15]. Furthermore, knowledge of MV is poor, even among intensivists [16]. Thus, the lack of educational methods is a problem in the current era of COVID-19, given the increasing demand for respiratory care, including MV management.

In Japan, the inadequacy of MV education has been noted, and there is no clearly established education curriculum on MV management for residents [17]. The planning and implementation of education on respiratory care, including MV, can be decided by each institution and supervisor. In addition, there are insufficient data on the needs of trainees, including residents, pertaining to MV management. Therefore, an effective and efficient educational strategy for MV management is required.

In this study, we aimed to develop a new educational strategy for MV management. We hypothesized that the use of flipped classrooms in MV education and the combination of multiple simulations could furthermore enhance educational effectiveness. Thus, the strategy consisted of e-learning and hands-on training with two kinds of simulators. The effectiveness of this educational strategy for residents was verified using a questionnaire.

## Methods

### Participants

In Japan, all residents have to complete a 2-year post-graduate residency in their primary subject, with rotations in other departments, such as internal medicine, surgery, emergency medicine, or anesthesia. The residents are also required to receive training for at least 3 months in the department of emergency or in the department of emergency and anesthesiology at their medical institute [17] within 2 years after graduating from a medical school. First- and second-year residents at Chiba University Hospital participated in our educational program in March 2022. Chiba University Hospital is a national university hospital with 850 beds. Among the fifty residents, twenty residents voluntarily participated in the program, and fourteen residents answered the pre-seminar and post-seminar survey. All the participants had undergone training in the department of emergency medicine and were in charge of patients requiring MV.

### Ethical approval

This study was approved by the Ethics Committee of Chiba University (approval no. 4106). The study database was anonymized, and the study complied with the requirements of the Japanese Ministry of Health, Labour and Welfare. Written informed consent was obtained from all participants when they answered the pre-seminar and post-seminar survey. The individual pictured in Fig 1 has provided written informed consent (as outlined in PLOS consent form) to publish their image alongside the manuscript All methods were performed in accordance with relevant guidelines and regulations.

### Preparation

The content of the educational program was based on the main findings of focus group interviews (FGIs) with respiratory physicians at our hospital. The semi-structured FGIs regarding the development of a new educational strategy for MV management were performed. FGIs were conducted by a physician researcher (HK), and the interview responses were recorded independently using an interview guide. Respiratory physicians were asked the following questions: (1) What difficulties have you experienced with respect to MV management? (2) What knowledge and abilities are necessary for proper MV management? (3) What knowledge and abilities do you lack to properly perform MV management? (4) What learning opportunities about MV management would you find useful? The interview guide was validated by the two researchers (HK and HT) before data collection. As a result, learning respiratory physiology was necessary and the program should be based on experientially learning. In addition, the following learning goal was set: to be capable of providing basic respiratory care during an outbreak of a respiratory infection.

### Educational program

For efficient learning, flipped classrooms were used, wherein e-learning was implemented first, followed by hands-on training. The two-hour educational program consisted of three sections: e-learning, hands-on training with a lung simulator (TTL$^{TM}$ Model Lung, Michigan Instruments, Talon Court SE, MI, USA), and hands-on training with a human patient simulator (SimMan 3G$^{TM}$, Laerdal Medical, Tanke Svilandsgate, Stavanger, Norway). Furthermore, this program was endorsed by three specialists: a respiratory medicine specialist (HK), a critical care specialist (TI), and a medical education specialist (SI).

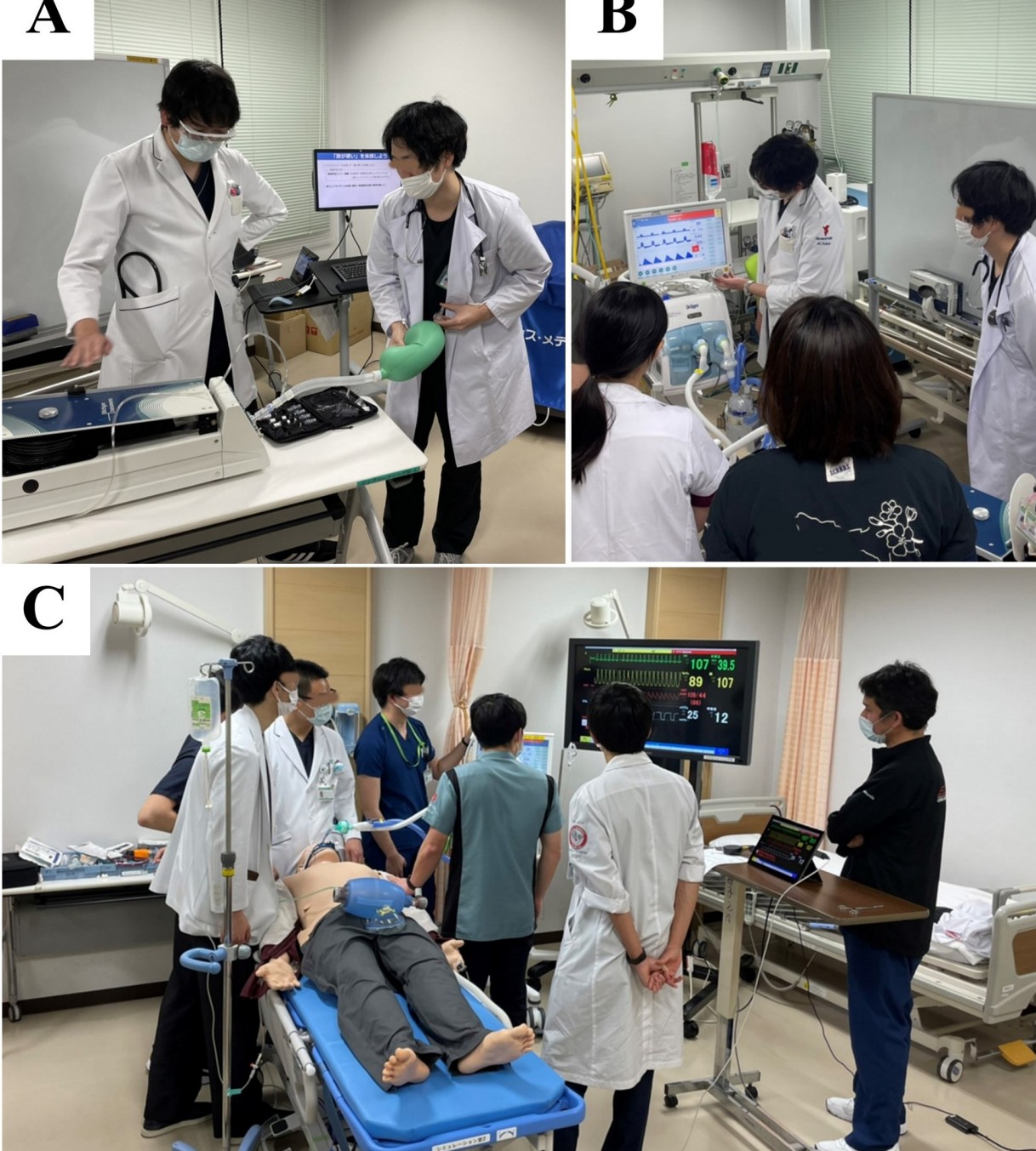

**Fig 1. Setting of the hands-on training.** (A, B) Images of hands-on training with a lung simulator; (C) image of hands-on training with a human patient simulator.

### e-learning for respiratory physiology and MV

The aim of the e-learning module was to impart knowledge on respiratory care, including MV. The 30-minute online module included lectures on respiratory physiology and MV, namely indications, introduction, troubleshooting, and weaning. The participants watched the

e-learning videos, which consist of descriptions with audio and slides, before starting the simulation-based training.

## Hands-on training with a lung simulator

The aim of the training with a lung simulator was to help the participants to learn about manual ventilation and MV techniques, to experientially learn (using visual and tactile stimuli) and understand the changes in lung compliance and airway resistance during manual and MV, and to be able to apply these changes to their ventilator settings. This activity also helped the participants apply the knowledge learned from the e-lectures. The session was conducted under the supervision of two respiratory physicians (KT, HK) and lasted approximately 45 minutes. For this activity, a lung simulator (TTL$^{TM}$ Model Lung) was connected to a Jackson Rees circuit and ventilator (Evita V500$^{TM}$, Dräger, Moislinger Allee, Lübeck, Germany) (Fig 1A and 1B).

## Hands-on training with a human patient simulator

The aim of the training with a human patient simulator was to help the participants obtain practical experience in assessing respiratory status and introducing MV. The simulation was conducted under the supervision of an intensivist (TI) and clinical engineer (YF). Severe respiratory failure due to infectious pneumonia was simulated using a human patient simulator (SimMan 3G$^{TM}$) that was connected to a breathing simulator (ASL 5000$^{TM}$, IngMar Medical, Pittsburgh, PA, USA). The learning activity was performed by teams of five-to-six participants at a time; they learned the process of MV administration, which included assessing the respiratory status, beginning manual ventilation, intubating, and adjusting the MV settings (Fig 1C).

## Evaluation of the effectiveness of the educational program

**Questionnaire survey.** Quantitative data were collected using questionnaires to evaluate the effectiveness of the education program on MV management. The participants' motivation and psychological burden pertaining to MV management were assessed using the questionnaire. Their understanding of terminologies related to MV management was also assessed. Before and after the e-learning and hands-on training, the residents responded to the following questionnaire items (Table 1): (1a) How motivated do you feel to learn about MV? (1b) How

**Table 1. Questionnaires for the participants.**

| Questions |
|---|
| 1a) How much do you feel a motivation to learn about MV? (1: Very weak to 5: Very strong) |
| 1b) How much do you feel a psychological burden of respiratory care required by patients needing MV management? (1: Very uncomfortable to 5: Very comfortable) |
| 1c) What do you find difficult about MV? (open-ended question) |
| 2) How do you understand the following terminology? (1: I have never heard it, 2: I have simply heard it, 3: I know it but cannot explain, 4: I can explain it). |
| a. Fraction of inspiratory oxygen ($F_IO_2$) |
| b. Positive end-expiratory pressure (PEEP) |
| c. Pressure support |
| d. Continuous positive airway pressure (CPAP) |
| e. Assist / control (A/C) |
| f. Airway pressure |
| g. Tidal volume |
| h. End tidal $CO_2$ ($ETCO_2$) |

much psychological burden of providing respiratory care to patients requiring MV management do you feel? (1c) What do you find difficult about MV? (2) How well do you understand the terminologies below? a. fraction of inspiratory oxygen ($F_IO_2$), b. positive end-expiratory pressure (PEEP), c. pressure support, d. continuous positive airway pressure (CPAP), e. assist/control (A/C), f. airway pressure, g. tidal volume, and h. end tidal $CO_2$ ($ETCO_2$).

Questions (1a) and (1b) were scored on a five-point Likert scale, with scores ranging from 1 (1a, very weak; 1b, very uncomfortable) to 5 (1a, very strong; 1b, very comfortable). Question (2) was scored on a four-point rating scale: 1 (I have never heard it), 2 (I have simply heard it), 3 (I know it but cannot explain it), and 4 (I can explain it). In addition, the participants reported their satisfaction level with the e-learning and hands-on training in a different questionnaire.

**Multiple-choice questions (MCQs).** The participants answered six MCQs on the evaluation of blood gas analysis, indications for MV, selection of MV mode depending on the patient's condition, method to modify hypoxemia/hypercapnia during MV, and weaning patients from MV. The MCQs were designed under the supervision of a respiratory medicine specialist (HK) and medical education specialist (SI).

## Data analysis

**Statistical analysis.** Quantitative data are expressed as mean ± standard deviation (SD), unless otherwise indicated. The Wilcoxon signed-rank test was used to compare the degree of psychological burden before and after MV education. A $p$-value $<0.05$ was considered statistically significant. All statistical analyses were performed using JMP 15.0 (Cary, North Carolina, USA).

## Results

### Demographics

The baseline characteristics of the study participants are presented in Table 2. Five (36%) first-year and nine (64%) second-year residents participated in this study. The participants were interested in internal medicine (n = 7, 50%) and surgery (n = 2, 14%). None of the residents were interested in emergency medicine or anesthesiology. All residents had the experience of being in charge of patients on MV.

**Table 2. Demographics of the participating residents.**

| Parameter | Results |
|---|---|
| Total, no. (%) | 14 (100) |
| Years after graduation, no. (%) | |
| First | 5 (36) |
| Second | 9 (64) |
| Department planned in future, no. (%) | |
| Internal medicine | 7 (50) |
| Surgery | 2 (14) |
| Emergency medicine / Anesthesiology | 0 (0) |
| Others | 5 (36) |
| Experience of mechanical ventilation, no. (%) | |
| None | 0 (0) |
| Observation only | 6 (43) |
| Experiences in patient care | 8 (57) |

### Pre- and post-seminar evaluations

In the pre-seminar evaluation, it was found that the participants experienced difficulties in introducing MV (9/14, 64%), troubleshooting in MV management (6/14, 43%), weaning patients from MV (5/14, 36%), and differentiating between models and manufacturers (2/14, 14%). Three (21%) and two (14%) residents, respectively, also noted a lack of experience in and knowledge of MV management.

The differences in the results of the pre- and post-seminar evaluations are shown in Table 3. The participants' motivation to learn about MV increased significantly after the program ($p = 0.016$). In addition, the understanding of terminologies improved, although there were no significant differences. In particular, the understanding of the following terms improved significantly: CPAP ($p = 0.016$), A/C ($p = 0.006$), and $ETCO_2$ ($p = 0.008$). However, the participants' psychological burden of MV management did not change significantly from the pre- to post-seminar evaluation ($p = 0.328$).

The differences in the pre- and post-seminar MCQs are shown in Table 4. The total MCQ scores improved significantly ($p = 0.003$). In particular, the accuracy of "method to modify hypoxemia" significantly improved (pre-seminar: 2/14; post-seminar 12/14; $p = 0.002$). The post-seminar scores on "assessment of blood gas analysis" and "indications for the MV" were better than the pre-seminar scores. The scores on the other questions tended to improve; however, there were no significant changes ("mode selection," from 5/14 [36%] to 7/14 [50%], $p = 0.688$; "method to modify hypercapnia," from 5/14 [36%] to 10/14 [71%], $p = 0.125$; and "weaning patient from MV," 7/14 (50%) to 10/14 [71%], $p = 0.453$).

## Discussion

The study revealed two important points related to MV management education for residents. First, the residents experienced difficulties in MV management and had a limited understanding of MV. Second, to our knowledge, this is the first study to evaluate a combination of e-learning videos and hands-on training with simulators (human patient and lung simulator) for MV training. In addition, this educational program increased the residents' motivation and knowledge on MV.

Although all residents had encountered patients who needed respiratory care, including MV, during their clinical training, four residents (29%) reported a lack of knowledge and experience in MV management in this study. For instance, in the pre-seminar questionnaire evaluation, some residents stated, "*I do not know the actual method of MV management*" and "*I do not have confidence in the area of MV management*." Hence, the residents may have had an insufficient understanding of MV management, indicating that the current on-the-job training on MV management may not fully meet the residents' needs, as also reported in previous studies [7]. In a study published in the United States in 2003, the residents did not receive sufficient education on MV during their training program and did not acquire the essential evidence-based knowledge necessary to provide effective care for MV [7]. Thus, educational interventions for MV management should be planned according to the residents' needs and readiness.

It is known that flipped classrooms are more effective than traditional classrooms for medical education [18], although it is unclear how e-learning affects MV management education. A previous study reported that clarification of learning objectives and adequate instructional time were correlated with satisfaction in MV training [7]. In addition, the combination of multiple methods based on the results of FGIs of respiratory physicians also improved the effectiveness of the program. In fact, all participants gave high ratings to the e-learning videos and the whole program, and two participants stated, "*I felt that I could learn efficiently*" and "*the*

**Table 3. Results of questionnaires for residents.**

| | Pre-seminar | Post-seminar | *P*-value |
|---|---|---|---|
| Residents' needs | | | |
| A motivation for learning mechanical ventilation, mean (standard deviation) | 4.4 (0.7) | 5.0 (0) | **0.016** |
| 1: very weak, no. (%) | 0 (0) | 0 (0) | |
| 2 | 0 (0) | 0 (0) | |
| 3 | 2 (14) | 0 (0) | |
| 4 | 5 (36) | 0 (0) | |
| 5: very strong, no. (%) | 7 (50) | 14 (100) | |
| A psychological burden to mechanical ventilation, mean (standard deviation) | 1.6 (0.9) | 2.2 (1.4) | 0.328 |
| 1: very uncomfortable, no. (%) | 9 (64) | 6 (43) | |
| 2 | 3 (21) | 4 (29) | |
| 3 | 1 (7) | 1 (7) | |
| 4 | 1 (7) | 1 (7) | |
| 5: very comfortable, no. (%) | 0 (0) | 2 (14) | |
| Understanding of terminologies | | | |
| Fraction of inspiratory oxygen, mean (standard deviation) | 3.6 (0.5) | 4.0 (0) | 0.063 |
| 1: I have never heard it, no. (%) | 0 (0) | 0 (0) | |
| 2: I have simply heard it, no. (%) | 0 (0) | 0 (0) | |
| 3: I know it but cannot explain, no. (%) | 5 (36) | 0 (0) | |
| 4: I can explain it, no. (%) | 9 (64) | 14 (100) | |
| Positive end-expiratory pressure, mean (standard deviation) | 3.6 (0.5) | 4.0 (0) | 0.063 |
| 1: I have never heard it, no. (%) | 0 (0) | 0 (0) | |
| 2: I have simply heard it, no. (%) | 0 (0) | 0 (0) | |
| 3: I know it but cannot explain, no. (%) | 5 (36) | 0 (0) | |
| 4: I can explain it, no. (%) | 9 (64) | 14 (100) | |
| Pressure support, mean (standard deviation) | 3.5 (0.6) | 3.9 (0.3) | 0.063 |
| 1: I have never heard it, no. (%) | 0 (0) | 0 (0) | |
| 2: I have simply heard it, no. (%) | 1 (7) | 0 (0) | |
| 3: I know it but cannot explain, no. (%) | 5 (36) | 1 (7) | |
| 4: I can explain it, no. (%) | 8 (57) | 13 (93) | |
| Continuous positive airway pressure, mean (standard deviation) | 3.3 (0.7) | 3.8 (0.4) | **0.016** |
| 1: I have never heard it, no. (%) | 0 (0) | 0 (0) | |
| 2: I have simply heard it, no. (%) | 2 (14) | 0 (0) | |
| 3: I know it but cannot explain, no. (%) | 6 (43) | 3 (21) | |
| 4: I can explain it, no. (%) | 6 (43) | 11 (79) | |
| Assist / control, mean (standard deviation) | 3.0 (0.7) | 3.7 (0.5) | **0.006** |
| 1: I have never heard it, no. (%) | 0 (0) | 0 (0) | |
| 2: I have simply heard it, no. (%) | 3 (21) | 0 (0) | |
| 3: I know it but cannot explain, no. (%) | 8 (57) | 4 (29) | |
| 4: I can explain it, no. (%) | 3 (21) | 10 (71) | |
| Airway pressure, mean (standard deviation) | 3.1 (0.6) | 3.6 (0.5) | 0.148 |
| 1: I have never heard it, no. (%) | 0 (0) | 0 (0) | |
| 2: I have simply heard it, no. (%) | 2 (14) | 0 (0) | |
| 3: I know it but cannot explain, no. (%) | 8 (57) | 6 (43) | |
| 4: I can explain it, no. (%) | 4 (29) | 8 (57) | |
| Tidal volume, mean (standard deviation) | 3.4 (0.5) | 3.9 (0.4) | 0.070 |
| 1: I have never heard it, no. (%) | 0 (0) | 0 (0) | |

(*Continued*)

**Table 3.** (Continued)

|  | Pre-seminar | Post-seminar | *P*-value |
|---|---|---|---|
| 2: I have simply heard it, no. (%) | 0 (0) | 0 (0) |  |
| 3: I know it but cannot explain, no. (%) | 8 (57) | 2 (14) |  |
| 4: I can explain it, no. (%) | 6 (43) | 12 (86) |  |
| End tidal $CO_2$ ($ETCO_2$), mean (standard deviation) | 3.1 (0.5) | 3.8 (0.4) | **0.008** |
| 1: I have never heard it, no. (%) | 0 (0) | 0 (0) |  |
| 2: I have simply heard it, no. (%) | 1 (7) | 0 (0) |  |
| 3: I know it but cannot explain, no. (%) | 10 (71) | 3 (21) |  |
| 4: I can explain it, no. (%) | 3 (21) | 11 (79) |  |

Bold characters indicate p<0.05.

*program was compact and comfortable with learning.*" The flipped classroom may promote the efficiency of education on MV management.

Simulators were introduced in the aviation industry in the 1920s and were first introduced in medical education in the 1960s [19]. Nowadays, simulation-based education is widely used in the field of medicine. Moreover, we found seven studies on simulation-based MV training for residents and fellows [8–14]. Six studies used human patient simulators and reported high satisfaction levels and improved test scores among the participants. These findings were consistent with the results of our study, suggesting that simulation-based education can be an effective strategy. In our study, the participants physically experienced the lung compliance and airway resistance during MV using their hands; this unique strategy helped them obtain a better understanding of positive pressure ventilation, pulmonary compliance, and airway resistance. A lung simulator can provide accurate representations of pulmonary mechanics and is used not only for training but also for testing and research and development of mechanical ventilators [20]. However, the educational effects of using a lung simulator have not yet been evaluated. A lung simulator can have advantages over a human patient simulator in that it allows the users to visualize the ventilation and airway pressure as well as physically experience low compliance and high airway pressure. A high-fidelity method, i.e., a human patient simulator, and a low-fidelity method, i.e., a lung simulator, may be complementary to each other. In addition, low-fidelity methods may be economically favorable and easier to implement.

Although the program examined in this study was shorter (2 hours) than that in previous studies (4–12 hours) [8–12], its educational effectiveness was suggested. The use of e-learning can help reduce the effort and time required to complete the education intervention for both

**Table 4. Results of multiple-choice questions.**

|  | Pre-seminar | Post-seminar | *P*-value |
|---|---|---|---|
| Total score*, mean (standard deviation) | 3.3 (1.1) | 4.6 (1.0) | **0.003** |
| Correction for each question, no. (%) |  |  |  |
| Evaluation of blood gas analysis | 13 (93) | 12 (86) | 1.000 |
| Indications for the mechanical ventilation | 14 (100) | 14 (100) | 1.000 |
| Mode selection | 5 (36) | 7 (50) | 0.688 |
| Method to modify hypoxemia | 2 (14) | 12 (86) | **0.002** |
| Method to modify hypercapnia | 5 (36) | 10 (71) | 0.125 |
| Weaning from mechanical ventilation | 7 (50) | 10 (71) | 0.453 |

Bold characters indicate p<0.05.

*Score ranges 0 to 6.

the educators and learners. Therefore, the results of our study may help medical schools establish an effective and efficient educational strategy for MV management. Although the participants' motivation and knowledge levels had improved, the psychological burden of MV management had not improved significantly. A previous study reported that 3 days of simulation-based education significantly increased the learners' confidence in MV management [9]. Hence, increased intervention time and increased use of simulators may be required to reduce the psychological burden of the residents.

This study has five limitations. First, this was a single-site and single-arm study with a small sample size; thus, the findings must be interpreted with caution. Second, although multiple methods were combined, the effectiveness of each one was not assessed separately. Third, the questionnaire that was used to assess the effectiveness of the intervention was not previously clinically validated. Fourth, long-term assessments were not conducted. Fifth, participant skills were not assessed by means of practical examinations.

In future research, the number of participants should be increased, and educational effects of the educational program, especially of a low-fidelity simulation with little evidence. In addition, the evaluation of educational effects could be improved, e.g., by using a simulation-based assessment [21]. Furthermore, the long-term educational effects should be evaluated with an attention to their association with the rotating departments before and after the intervention. When studies are conducted to evaluate these aspects, it would be beneficial to set a control group.

## Conclusions

Our new educational strategy for training residents on MV could increase residents' motivation to learn about respiratory care and improve their knowledge of MV management in a short time. In particular, the flipped classroom, which combined e-learning and hands-on training (with high- and low-fidelity simulators), increased the efficiency of our educational strategy. Our educational strategy was implemented in the pilot study phase of a single-arm study with a small sample size. Further evaluations are needed to establish an effective and practical educational strategy for MV management.

## Supporting information

**S1 File.**
(XLSX)

## Author Contributions

**Conceptualization:** Kenichiro Takeda, Hajime Kasai, Hiroshi Tajima, Shoichi Ito.

**Data curation:** Kenichiro Takeda.

**Formal analysis:** Kenichiro Takeda.

**Investigation:** Kenichiro Takeda, Yutaka Furukawa, Taro Imaeda.

**Methodology:** Kenichiro Takeda, Hajime Kasai, Taro Imaeda, Shoichi Ito.

**Project administration:** Kenichiro Takeda.

**Resources:** Kenichiro Takeda.

**Supervision:** Hajime Kasai, Hiroshi Tajima, Yutaka Furukawa, Taro Imaeda, Takuji Suzuki, Shoichi Ito.

**Validation:** Hajime Kasai, Hiroshi Tajima.

**Writing – original draft:** Kenichiro Takeda.

**Writing – review & editing:** Hajime Kasai, Hiroshi Tajima, Yutaka Furukawa, Taro Imaeda, Takuji Suzuki, Shoichi Ito.

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
