## [Decision Letter · Decision Letter 0]

17 May 2023

PONE-D-23-05208Mixed-methods education of mechanical ventilation for residents in the era of the COVID-19 pandemic: preliminary interventional studyPLOS ONE

Dear Dr. Takeda,

Thank you for submitting your manuscript to PLOS ONE. After careful consideration, we feel that it has merit but does not fully meet PLOS ONE’s publication criteria as it currently stands. Therefore, we invite you to submit a revised version of the manuscript that addresses the points raised during the review process.

ACADEMIC EDITOR: Both Reviewers, with experience in education on mechanical ventilation, underlined the interest on the paper. However, the small sample size and the lack of external validity must be strongly commented. I may suggest to insert that the study is pilot and more data are needed.

We look forward to receiving your revised manuscript.

Kind regards,

Andrea Cortegiani, M.D.

Academic Editor

PLOS ONE

2. Please provide additional details regarding participant consent. In the ethics statement in the Methods and online submission information, please ensure that you have specified (1) whether consent was informed and (2) what type you obtained (for instance, written or verbal, and if verbal, how it was documented and witnessed). If your study included minors, state whether you obtained consent from parents or guardians. If the need for consent was waived by the ethics committee, please include this information. If you are reporting a retrospective study of medical records or archived samples, please ensure that you have discussed whether all data were fully anonymized before you accessed them and/or whether the IRB or ethics committee waived the requirement for informed consent. If patients provided informed written consent to have data from their medical records used in research, please include this information.

Reviewers' comments:

Reviewer's Responses to Questions

**Comments to the Author**

1. Is the manuscript technically sound, and do the data support the conclusions?

Reviewer #1: Yes

Reviewer #2: Yes

2. Has the statistical analysis been performed appropriately and rigorously? 

Reviewer #1: Yes

Reviewer #2: Yes

3. Have the authors made all data underlying the findings in their manuscript fully available?

Reviewer #1: Yes

Reviewer #2: Yes

4. Is the manuscript presented in an intelligible fashion and written in standard English?

Reviewer #1: Yes

Reviewer #2: Yes

5. Review Comments to the Author

Reviewer #1: Every technical field has been respected.

Using flipped classroom e-learning together with lung simulators and human patient simulators is certainly a new and very useful method to prepare the next generation of physicians to deal with MV in real life and in emergency situations (as covid 19 pandemic), so these types of studies are not only worthwhile but necessary. On the other hand such a small sample of residents is not enough to assess the effectiveness of the training, and the duration of the training is too short. But those two objections are correctly highlighted at the end of the paper. If the ultimate goal of this studies is to start creating a standardized learning method this is a good starting point.

Reviewer #2: Dear Editor and Authors,

Thank you for the opportunity to revise the manuscript titled “Mixed-methods education of mechanical ventilation for residents in the era of the COVID-19 pandemic: preliminary interventional study”.

The manuscript aims to investigate preliminary educational strategies for mechanical ventilation management in a small cohort of residents through the use of flipped classrooms and two different types of simulators.

The design of the study is interventional. The authors found that the methods that were used increased residents motivation to learn about respiratory care and improved thei knowledge about mechanical ventilation.

The article is well written, methods are clearly stated, and the results are well expressed.

In my opinion, though, minor revision is needed:

- Page 3 lines 54-57: the sentence is too long, please rephrase accordingly.

- Page 4 line 66: In this section the authors go through some of the literature regarding simulation and mechanical ventilation teaching. The authors may want to see also the following literature: DOI: 10.34197/ats-scholar.2020-0023OC; DOI:10.1016/j.pulmoe.2022.05.007

- Page 9 lines 169-171: Maybe this sentence could be better expressed in order to clarify the message.

- Page 15 line 245: Although very interesting the discussion results in being too long and with several repetitions regarding the findings. The paper would benefit from a more concise approach in this section.

- Page 18 line 313: The limitations are clearly stated. What about the lack of a control group? Would the authors think it should be mentioned as a limitation and as a cue to improve future research on the topic?

6. PLOS authors have the option to publish the peer review history of their article (what does this mean?). If published, this will include your full peer review and any attached files.

Reviewer #1: **Yes: **Marta Milazzo

Reviewer #2: No

---

## [Author Response · Author response to Decision Letter 0]

11 Jun 2023

June 9, 2023

Andrea Cortegiani, M.D.

Academic Editor

PLOS ONE

Dear Editor: 

Thank you for your email dated May 18, 2023, regarding our manuscript. We are grateful for the opportunity to revise our manuscript. Based on your comments, we wish to resubmit our manuscript titled “Mixed-methods education of mechanical ventilation for residents in the era of the COVID-19 pandemic: preliminary interventional study.” The manuscript number is PONE-D-23-05208.

We thank you and the reviewers for your thoughtful suggestions and insights. The manuscript has benefited from these insightful suggestions. I look forward to working with you and the reviewers to move this manuscript closer to publication in PLOS ONE.

The manuscript has been rechecked, and the necessary changes have been made in accordance with the reviewers’ suggestions. The responses to all comments have been prepared and given below. The changes made to the main manuscript are highlighted in yellow. We have also added in the revised manuscript that all participants provided written informed consent.

We believe that this version of the manuscript has addressed all the reviewers’ questions. We hope that you will now find it suitable for publication in PLOS ONE.

Thank you for your consideration. I look forward to hearing from you.

Sincerely,

Kenichiro Takeda

Department of Respirology Graduate School of Medicine, Chiba University, Chiba 260-8670, Japan Telephone: +81-43-222-7171 Ext. 72851/ Fax: +81-43-226-2176 Email: k.takeda@chiba-u.jp

 

Response to Academic Editor’s comments

General Comment:

Both Reviewers, with experience in education on mechanical ventilation, underlined the interest on the paper. However, the small sample size and the lack of external validity must be strongly commented. I may suggest to insert that the study is pilot and more data are needed.

Response: 

We would like to thank you for taking the time and effort to review our manuscript and providing us the opportunity to revise our manuscript. We did not emphasize the limitations of our study in the manuscript. 

We mentioned that our study was a preliminary study in the title of the original manuscript, and in the discussion, we stated that the small sample size was a limitation. As you suggested, we have now added that our study was a pilot study in the “Conclusion” section. 

We have marked the relevant changes in red font in this letter.

Original: Page 19, Conclusion

We implemented a new educational strategy for training residents on MV. The seminar increased their motivation to learn about respiratory care and improved their knowledge of MV management in a short time. In particular, the flipped classroom, which combined e-learning and hands-on training (with high- and low-fidelity simulators), increased the efficiency of our educational strategy. However, the effectiveness of the lung simulator should be further evaluated to establish a more effective and practical educational strategy for MV management.

Revised: 

Our new educational strategy for training residents on MV could increase residents’ motivation to learn about respiratory care and improve their knowledge of MV management in a short time. In particular, the flipped classroom, which combined e-learning and hands-on training (with high- and low-fidelity simulators), increased the efficiency of our educational strategy. Our educational strategy was implemented in the pilot study phase of a single-arm study with a small sample size. Further evaluations are needed to establish an effective and practical educational strategy for MV management.

 

Response to Reviewer #1’s comments

General Comment:

Every technical field has been respected.

Using flipped classroom e-learning together with lung simulators and human patient simulators is certainly a new and very useful method to prepare the next generation of physicians to deal with MV in real life and in emergency situations (as covid 19 pandemic), so these types of studies are not only worthwhile but necessary. On the other hand such a small sample of residents is not enough to assess the effectiveness of the training, and the duration of the training is too short. But those two objections are correctly highlighted at the end of the paper. If the ultimate goal of this studies is to start creating a standardized learning method this is a good starting point.

Response: 

We would like to thank you for taking the time and effort to review our manuscript and for your valuable comments. 

Small sample size and lack of assessment of the long-term educational effect are the limitations of our study. We are planning to continue to accumulate more data by increasing the number of seminars and conducting them at multiple institutions for a wide variety of health care professionals. 

In the revised manuscript, we have added the limitations of this study and future plans, based on the comments from other reviewers.

 

Response to Reviewer #2’s comments

General Comment:

Thank you for the opportunity to revise the manuscript titled “Mixed-methods education of mechanical ventilation for residents in the era of the COVID-19 pandemic: preliminary interventional study”.

The manuscript aims to investigate preliminary educational strategies for mechanical ventilation management in a small cohort of residents through the use of flipped classrooms and two different types of simulators.

The design of the study is interventional. The authors found that the methods that were used increased residents’ motivation to learn about respiratory care and improved their knowledge about mechanical ventilation.

The article is well written, methods are clearly stated, and the results are well expressed.

In my opinion, though, minor revision is needed:

Response:

We would like to thank you for taking the time and effort to review our manuscript and for your valuable comments and suggestions, which have considerably helped us improve our manuscript. 

We have answered your points below and hope that our responses and revisions have addressed all your comments.

We have highlighted the respective changes in yellow to make them easily identifiable.

Comment #1

- Page 3 lines 54-57: the sentence is too long, please rephrase accordingly.

Response:

We thank you for your suggestion. The text was revised accordingly.

Original: Page 3 lines 54-57, 1st paragraph of Introduction

In hindsight, it is evident that not only COVID-19 but also other respiratory infections, such as the severe acute respiratory syndrome and Middle East Respiratory Syndrome, which caused severe respiratory failure, have broken out regularly throughout the world or in specific regions.

Revised: In hindsight, other types of respiratory infections that cause severe respiratory failure have broken out regularly throughout the world or in specific regions. 

Comment #2

- Page 4 line 66: In this section the authors go through some of the literature regarding simulation and mechanical ventilation teaching. The authors may want to see also the following literature: DOI: 10.34197/ats-scholar.2020-0023OC; DOI:10.1016/j.pulmoe.2022.05.007

Response: 

We would like to thank you for your valuable recommendations. 

We have added the two papers you suggested to our reference list. Both papers reported the effect of simulation-based education on mechanical ventilation and were useful references for us. The text has been revised accordingly.

Original: Page 17 lines 281-283, 4th paragraph of Discussion

Moreover, we found five studies of simulation-based MV training for residents and fellows [8-12]. All of these studies used human patient simulators and reported high satisfaction and improved test scores among the participants.

Revised: Moreover, we found seven studies on simulation-based MV training for residents and fellows [8-14]. Six studies used human patient simulators and reported high satisfaction levels and improved test scores among the participants.

Comment #3

- Page 9 lines 169-171: Maybe this sentence could be better expressed in order to clarify the message.

Response: Thank you for your suggestion. We have attempted to improve this sentence, and it has been reviewed by native English speakers.

Original: Page 9 lines 169-171 Method, Evaluation of the effectiveness of the educational program, Questionnaire survey

The questionnaire was designed to assess the participants’ motivation to learn about MV and the psychological burden of respiratory care required by patients requiring MV management.

Revised:

The participants’ motivation and psychological burden pertaining to MV management were assessed using the questionnaire. 

Comment #4

- Page 15 line 245: Although very interesting the discussion results in being too long and with several repetitions regarding the findings. The paper would benefit from a more concise approach in this section.

Response:

Thank you for raising this concern. We would like to apologize for the redundancy. 

Accordingly, we have deleted repetitive sentences as much as possible and clarified the related sentences.

Original: Page 15 lines 246-253, 1st paragraph of Discussion

The study revealed two important points related to MV management education for residents. First, the residents subjectively found it difficult to manage patients requiring MV. They also did not sufficiently understand how each parameter of positive pressure ventilation affected breathing. Second, to our knowledge, this is the first study to evaluate a combination of e-learning videos and hands-on training with simulators (human patient and lung) for MV training. Our educational program increased the residents’ level of motivation to learn about MV and knowledge of managing MV. Therefore, the data obtained in this study would be valuable for establishing new educational strategies in this regard.

Revised:

The study revealed two important points related to MV management education for residents. First, the residents experienced difficulties in MV management and had a limited understanding of MV. Second, to our knowledge, this is the first study to evaluate a combination of e-learning videos and hands-on training with simulators (human patient and lung simulator) for MV training. In addition, this educational program increased the residents’ motivation and knowledge on MV. 

Original: Page 15 lines 254-257, 2nd paragraph of Discussion

The residents’ experiences and needs related to MV management were surveyed using questionnaires. Although all residents had encountered patients who needed respiratory care, including MV, during their clinical training, four residents (29%) reported a lack of knowledge and experience in MV management.

Revised:

Although all residents had encountered patients who needed respiratory care, including MV, during their clinical training, four residents (29%) reported a lack of knowledge and experience in MV management in this study.

Original: Page 16 lines 263-265, 2nd paragraph of Discussion

approximately half of the residents stated that their training programs did not spend enough time on MV, and their test scores showed that they did not gain essential evidence-based knowledge needed to provide effective care to patients who require MV [7].

Revised:

…the residents did not receive sufficient education on MV during their training program and did not acquire the essential evidence-based knowledge necessary to provide effective care for MV [7].

Deleted: Page 16 lines 271-273, 3rd paragraph of Discussion

In our novel educational strategy, we presented the residents with the learning objectives before the program and provided instructions via a flipped classroom because 

Deleted: Page 17 lines 293-295, 4th paragraph of Discussion

The effectiveness the lung simulator could not be fully evaluated in our study because multiple methods were combined in our educational strategy. However,

Deleted: Page 18 lines 302-304, 5th Paragraph of Discussion

A combination of multiple educational methods may improve the efficiency of MV management education. In addition,

Comment #5

- Page 18 line 313: The limitations are clearly stated. What about the lack of a control group? Would the authors think it should be mentioned as a limitation and as a cue to improve future research on the topic?

Response:

We would like to thank you for your careful review. This study was a pilot, single-arm study with a small number of participants. Thus, our study assessed the limited and short-term effects of a single educational intervention. In the future, we will consider setting a control group when conducting multiple interventions or assessing the long-term educational effects. 

Accordingly, we have revised the related sentences as follows.

Original: Page 18 line 313-314, 6th paragraph of Discussion

This study has five limitations. First, this was a single-site study with a small sample size; thus, the findings must be interpreted with caution. 

Revised: 

This study has five limitations. First, this was a single-site and single-arm study with a small sample size; thus, the findings must be interpreted with caution.

Added: 7th paragraph of Discussion

When studies are conducted to evaluate these aspects, it would be beneficial to set a control group.

---

## [Editor Report · Decision Letter 1]

15 Jun 2023

Mixed-methods education of mechanical ventilation for residents in the era of the COVID-19 pandemic: preliminary interventional study

PONE-D-23-05208R1

Dear Dr. Takeda,

We’re pleased to inform you that your manuscript has been judged scientifically suitable for publication and will be formally accepted for publication once it meets all outstanding technical requirements.

Kind regards,

Andrea Cortegiani, M.D.

Academic Editor

PLOS ONE
---

## [Editor Report · Acceptance letter]

6 Jul 2023

PONE-D-23-05208R1 

Mixed-methods education of mechanical ventilation for residents in the era of the COVID-19 pandemic: preliminary interventional study 

Dear Dr. Takeda:

I'm pleased to inform you that your manuscript has been deemed suitable for publication in PLOS ONE. Congratulations! Your manuscript is now with our production department. 

Kind regards, 

on behalf of

Dr. Andrea Cortegiani 

Academic Editor

PLOS ONE